# Enhancing the Growth of Chili Plants and Soil Health: Synergistic Effects of Coconut Shell Biochar and *Bacillus* sp. Strain Ya-1 on Rhizosphere Microecology and Plant Metabolism

**DOI:** 10.3390/ijms252011231

**Published:** 2024-10-18

**Authors:** Shimeng Tan, Bao Wang, Qian Yun, Wanrong Yan, Tongbin Xiao, Zhixiang Zhao

**Affiliations:** 1Key Laboratory of Plant Disease and Pest Control of Hainan Province, Institute of Plant Protection (Research Center of Quality Safety and Standards for Agricultural Products of Hainan Academy of Agricultural Sciences), Hainan Academy of Agricultural Sciences, Haikou 571100, China; tshmeng@126.com (S.T.); wangbao@hnaas.org.cn (B.W.); yanwanrong818@163.com (W.Y.);; 2Key Laboratory of Green Prevention and Control of Tropical Plant Diseases and Pests Ministry of Education, Hainan University, Haikou 571100, China

**Keywords:** biochar application, *Bacillus* inoculation, rhizosphere microbiology, soil enhancement, *Capsicum annuum*

## Abstract

To mitigate soil degradation and decrease dependency on chemical inputs in agriculture, this study examined the joint effects of coconut shell biochar and *Bacillus* strain Ya-1 on soil fertility, rhizosphere bacterial communities, and the growth of chili (*Capsicum annuum* L.). A controlled pot experiment with four treatments was conducted: control (CK), biochar only (C), *Bacillus* strain Ya-1 only (B), and a combination of both (BC). The BC treatment significantly enhanced the soil carbon and available phosphorus contents by approximately 20% and the soil nitrogen content and pH by 18% and 0.3 units, respectively, compared to the control. It also increased microbial biomass carbon and nitrogen by 25% and 30%, respectively, indicating improved soil microbial diversity as shown by the highest Pielou evenness index and Shannon index values. The combined application of biochar and the Ya-1 strain resulted in a 15% increase in chili plant height and a 40% improvement in root dehydrogenase activity, suggesting enhanced nutrient uptake and metabolism. Metabolic profiling showed shifts in stress response and nutrient assimilation under different treatments. Collectively, these results indicate the potential of biochar and microbial inoculants to significantly promote soil and plant health, providing a sustainable strategy to improve agricultural productivity and reduce reliance on chemical inputs.

## 1. Introduction

Chili (*Capsicum annuum* L.) is a globally important vegetable crop in the agriculture sector, both economically and as a cooking staple, particularly in Hainan, China, which is a major supplier of off–season vegetables. The warm and humid climate dominating this region, however, increases the prevalence of some soil–borne pathogens, such as *Meloidogyne enterolobii* [1], *Fusarium oxysporum* [2], and *Pseudomonas solanacearum* [3], by extending their lifecycle and accelerating their activity, which consequently pose a severe threat to the production of peppers, significantly reducing their yield and quality.

The pressing environmental and health issues associated with the extensive use of chemical pesticides call for the urgent need to adopt sustainable agricultural practices. In this context, there is growing popularity in the application of biological control agents (BCAs), such as *Pseudomonas fluorescens* and *Bacillus subtilis* [4], as well as soil amendments like biochar [5]. BCAs protect plants against pathogens through mechanisms such as triggering antibacterial activity, inducing systemic resistance [6], and hyperparasitism [7], offering an eco-friendly alternative to synthetic chemicals. Meanwhile, the mediation of the interactions between microbes in the rhizosphere environment and plants by a variety of secondary metabolites and signaling molecules has been suggested by some studies that have investigated the effects of rhizosphere microecology on plant immunity. Song et al. [8] found that the generation of ROS mediated by FER can regulate the levels of plant-beneficial *Pseudomonas* spp. in the Arabidopsis rhizosphere microbiome. The *Pseudomonas*-specific induction of the expression of PSKR1 in Arabidopsis roots further demonstrated the role of salicylic acid in regulating the colonization ability and growth of *Pseudomonas*, indicating the ways through which the plant’s immunity is boosted in coordination with positive functions of the microbiome.

With unique properties such as high porosity, sufficiently large surface area, and nutrient retention capacity, biochar is renowned for its ability to improve soil structure, which supports the growth of beneficial microbes and enhances plant health and productivity [9]. Beyond its ample carbon and elemental contributions, biochar fortifies aggregation in finer-textured soils, ameliorating the capillary structure within compacted soils, and, consequently, bestows significant functions related to this structure, including water retention and the enhancement of microbial communities [10]. In this context, the synergistic application of biochar and BCAs has been observed to amplify the enhancement effects on farmland soil health and plant disease resistance, surpassing the benefits of biochar alone [11]. A readily observable rationale is the substantial retention of active functional groups, such as hydroxyl, carboxyl, sulfonic, and amino groups, on the biochar surface, which are highly effective for microbial cell adhesion and proliferation, facilitating biofilm formation and providing an excellent habitat and refuge for microbes [12]. On the other hand, research into rhizosphere microbial communities has revealed that the addition of biochar can enrich the basic BCA community, encompassing Pseudomonas, *Bacillus*, and Streptomyces, while also supporting the active manifestation of rhizosphere metabolic enzymes [13]. By stabilizing and supporting the functionality of microbial communities, biochar, when used as a carrier for microbial inoculants, has been demonstrated to enhance the persistence, survival, and colonization of inoculated microbes in soil and on plant roots, which is clearly advantageous in overcoming the disadvantages of low functional efficiency and short utility periods of microbial inoculants. The combined use of biochar and microbial inoculants has proven to be an effective strategy for controlling a variety of plant pathogens, highlighting its potential contribution to integrated disease management in agriculture [14].

In this study, we conducted a comparative analysis between the sole application and the combined application of coconut shell biochar and *Bacillus* strain Ya-1 in pot experiments, integrating the analysis of rhizosphere microbial diversity and plant metabolism to explore potential synergistic interactions and, more importantly, to analyze the impact of these measures on plant metabolism. The pot experiment design, conducted under controlled conditions, allows for the precise manipulation and monitoring of both environmental variables and treatment effects. Through this design, we aim to elucidate the precise effects of these combined application measures on plant immunity before further pursuing the microbial agents carried by biochar. This experimental setup provides profound insights into the interactions established between biochar, *Bacillus* strain Ya-1, and the plant, laying the groundwork for the potential future co-application of biochar and microorganisms under field conditions.

## 2. Results

### 2.1. Combined Effects of Biochar and Strain Ya-1 on the Soil and Rhizosphere Bacterial Communities

#### 2.1.1. Soil Physicochemical Properties

Figure 1 shows the results of a comparative analysis of the soil’s physicochemical properties in different treatments. The control group (CK) was used to establish a baseline for all evaluated parameters. Significant increases in the carbon (C) and available phosphorus (AP) contents were recorded with biochar (C) addition, while other parameters, such as the nitrogen (N) content and microbial biomass carbon (C), moderately changed. The sole application of strain Ya-1 (B), however, notably increased the microbial biomass N and P contents, indicating the enhancement of microbial activity, and also significantly improved the water content. The biochar treatment in combination with Ya-1 (BC) resulted in the maximum contents of total P and C as well as available P and potassium (K). Moreover, compared to the treatment of plants with only Ya-1 (B), a noticeable increase was shown in microbial C and P and water content with BC, which indicates the synergistic effect of the combined use of biochar and Ya-1.

The heatmap shows the effects of different treatments, including the control (CK), biochar addition alone (C), Ya-1 addition alone (B), and biochar combined with Ya-1 (BC), on soil physicochemical properties. The parameters evaluated included the potassium (K), nitrogen (N), phosphorus (P), and carbon (C) contents; microbial biomass N; available K, N, and P contents; microbial biomass C and P; pH; and water content. The data were normalized and color coded, with a yellow color indicating higher contents, and purple indicating lower contents.

#### 2.1.2. The Rhizosphere Bacterial Diversity of Chili Plants

The boxplots in Figure 2 show the effects of different treatments on bacterial diversity indices. In the BC treatment, significantly higher values of Pielou’s evenness index compared to those of B and CK were obtained, which demonstrated a more even distribution of the bacterial community. The intermediate evenness of the bacterial community was observed for the C treatment, which was on par with that for BC. In terms of species richness, the plants exposed to both BC and C treatments exhibited significantly higher values than those under B and CK, and plants grown under the BC treatment achieved the highest richness, suggesting that biochar could promote bacterial diversity. Similarly, the BC group, in tandem with the C group, exhibited the highest Shannon diversity index, indicating that biochar induced a comprehensive enhancement in bacterial diversity. It is important to highlight that the integration of Ya-1 with biochar did not create a significant divergence in diversity indices compared to the biochar-only treatment. Furthermore, the introduction of Ya-1 did not upset the community equilibrium, even in large quantities, suggesting that biochar plays a role in the stabilization of microbial communities.

### 2.2. Combined Effects of Biochar and Strain Ya-1 on the Growth of Chili Plants

The effects of biochar and the strain Ya-1 on the growth of chili plants were evaluated by measuring the total height of the plant, including the root and stem lengths, in different treatments, as shown in Figure 3 and Figure 4. Figure 3E demonstrates that the control group (CK) plants exhibited a mean total height of 375 mm, with average values of 150 mm and 225 mm for the roots and stems, respectively. When applied alone, biochar (C) caused a significant increase in the total plant height, averaging 475 mm, with the roots and stems having lengths of 175 mm and 300 mm, respectively. In contrast, a total plant height similar to that of the control group was shown when Ya-1 (B) was applied alone, suggesting that plant growth was negligibly influenced by the bacterial strain. However, the co-application of biochar and Ya-1 (BC) exerted the most pronounced effect, with average values of over 230 mm and 215 mm for the stem length and root length of the longest plant, respectively, which points out their synergistic influence on plant growth.

The leaf surface area, an important determinant of photosynthetic capacity in plants, displayed variations among different treatment groups. A median leaf surface area of approximately 1200 mm^2^ was achieved for the control plants (CK), within the range of 1000 mm^2^ to 1600 mm^2^ (Figure 3F). The biochar treatment (C) caused a reduction of about 1000 mm^2^ in the median leaf surface area, with a narrower range of 800 mm^2^ to 1200 mm^2^. Conversely, an increase in the median leaf area to approximately 1400 mm^2^, within a broader range of 800 mm^2^–1600 mm^2^, was seen with the sole application of Ya-1 (B). The leaf surface area underwent the highest variation, with a median of 1200 mm^2^ and a range of 800 mm^2^–1600 mm^2^, as a result of the co-treatment of biochar and Ya-1 (BC).

The dehydrogenase activity of plant roots, which is a biological indicator of root health, was assessed in all treatment groups, as presented in Figure 3G. The plant roots in the control group (CK) displayed an average dehydrogenase activity of 0.045 mg/(g·h). A slight enhancement of this activity (an average of 0.048 mg/(g·h)) was recorded for the biochar treatment (C), while the addition of Ya-1 (B) slightly decreased the dehydrogenase activity, with a mean value of 0.042 mg/(g·h). Most importantly, when biochar and Ya-1 (BC) were applied in combination, significant increases in the dehydrogenase activity of the roots, with an average of 0.065 mg/(g·h), resulted, indicating that together they could promote the metabolic activity in plant roots.

The results revealed a significant improvement in the growth parameters of chili plants with the combined application of biochar and Ya-1. In the BC treatment, not only was the maximum plant height achieved but also the metabolic activity of roots was enhanced. This indicated that it was a particularly effective treatment and was manifested by the elevated dehydrogenase activity of roots. These findings suggest that the combined use of biochar and Ya-1 could synergistically promote the physical and metabolic development of chili plants, potentially leading to a boost in plant health and productivity.

### 2.3. Effects of the Co-Application of Biochar and Ya-1 on Metabolic Changes in Chili Plants

The comparative metabolomic analysis of chili plants in different treatments, including biochar (C), *Bacillus* strain (B), and biochar–*Bacillus* combination (BC), revealed significant metabolic shifts. Figure 4 represents the key findings from the comparative metabolomic analysis.

Figure 4A presents the volcano plot of differential metabolites between the C and BC treatments, which exhibited a stark difference, and 7282 metabolites were upregulated, while 2816 metabolites were downregulated in the BC treatment compared to those in C. Metabolites with a *p*-value less than 0.05 were marked as statistically significant, and red and blue represent the upregulated and downregulated metabolites, respectively. Furthermore, metabolites with high variable importance in the projection (VIP) values were prominently highlighted. Similarly, the volcano plot illustrating the differential metabolites between the B and BC treatments, with 4424 upregulated metabolites but 6384 downregulated metabolites in the treatment BC compared to those in B, is shown in Figure 4B.

The matchstick diagrams represent the top differential metabolites. The top 10 differential metabolites between the C and BC treatments are shown in Figure 4C, with the significant upregulation of N-methylglycine and dodecanoylcarnitine, whereas pristimerin and caffeic acid glycoside exhibited a notable downregulation. The length of the lines in the plot corresponds to the magnitude of fold change, and the red color indicates upregulation, while blue denotes downregulation. Likewise, Figure 4D demonstrates the key upregulated differential metabolites identified, such as quercetin 3-O-(6-O-malonyl-*β*-D-glucoside) and N-methylglycine, and downregulated metabolites, including ginsenoside F1 and allopurinol, between the B and BC treatments.

The pathway enrichment analysis further revealed the induction of metabolic changes by different treatments. As seen in Figure 4E, a heatmap of the KEGG pathway enrichment analysis of differential metabolites between the C and BC treatment groups represents significantly enriched pathways associated with amino acid metabolism and the KEGG pathway of biosynthesis of various other secondary metabolites. In the heatmap, the intensity of the color reflects the degree of enrichment, with a darker shade indicating a higher level of significance. Similarly, Figure 4F illustrates the enrichment analysis of KEGG pathways related to differential metabolites identified between the *Bacillus* treatment (B) and the Biochar–*Bacillus* treatment (BC). This heatmap depicts several significantly enriched metabolic pathways, particularly “amino sugar and nucleotide sugar metabolism” and “isoquinoline alkaloid biosynthesis”.

Treemaps provide a visual representation of statistically significant pathways, highlighting the relative contribution of these pathways to the overall metabolic change in plants. In the treemap, the size of each rectangle corresponds to the significant enrichment of the KEGG pathway, with larger rectangles indicating greater impacts, and larger and darker squares denoting more significantly enriched pathways. Figure 4G shows the tree map, which offers the representation of the KEGG pathway enrichment analysis of differential metabolites between the treatments C and BC, emphasizing the relative importance of the two pathways of alanine, aspartate, and glutamate metabolism and isoquinoline alkaloid biosynthesis. Similarly, a tree map of the KEGG pathway enrichment analysis of differential metabolites between the B and BC treatments is presented in Figure 4H. Notably, the largest and darkest rectangles in the treemap represent pathways such as “amino sugar and nucleotide sugar metabolism” and “isoquinoline alkaloid biosynthesis”, underscoring their importance to the induction of metabolic shifts by the BC treatment.

Substantial metabolic changes were observed in chili plants when biochar was integrated with *Bacillus* (BC) compared to those in the individual treatments of biochar (C) and *Bacillus* (B). The BC treatment triggered significant alterations in key metabolites and metabolic pathways, suggesting the synergistic effect of the combined application of biochar and *Bacillus* on the metabolism of chili plants, which could be a promising strategy to improve metabolic processes, potentially promoting plant growth and stress resilience.

### 2.4. Integrative Analysis of the Effects of the Co-Application of Biochar and Bacillus

The integrative analysis investigates the biochemical and microbial processes taking place in chili plants treated with biochar combined with *Bacillus* (BC) compared to those under the sole applications of C and B. Pearson correlation heatmaps (Figure 5) revealed how plant metabolites and soil microbial diversity were influenced by the BC treatment, highlighting the synergistic effects of biochar and *Bacillus* on plant growth and soil health. Some plant growth-promoting metabolites, such as “3,5-Dimethoxycinnamic acid” in Group B, exhibited significant positive correlations with key growth parameters like the root length and leaf area, which the biochar–*Bacillus* combination treatment effectively mitigated. Similarly, in Group C, the treatment notably potentiated the beneficial effects of “4-Phenylbutanoic Acid” on these traits. This suggests that the combined application of biochar and bacterial strains not only enhances the production of specific beneficial metabolites but also has the potential to improve overall plant growth vigor and mitigate some of the negative impacts of certain metabolites on plant growth. Additionally, the correlations between the Shannon index and metabolites such as “pyroglutamic acid” and “5-(3-Buten-1-ynyl)-2,2′-bithiophene” indicated an increase in the soil bacterial diversity by the BC treatment, implying that biochar and *Bacillus* could together improve the soil environment, which supports better plant growth.

The heatmap (Figure 6) shows the coefficients of the Pearson correlation between differential metabolites and growth parameters for the treatment groups C versus BC (left) and B versus BC (right). The intensity of the color of each cell reflects the magnitude of the correlation coefficient, with blue indicating a negative correlation, red denoting a positive correlation, and white representing the non-significant correlation. The absolute value of the coefficient close to 1 indicates a strong linear relationship between the variables. Correlations between the top 20 metabolites and the leaf area, root vitality (indicated by dehydrogenase activity), and morphological traits such as root and stem lengths, were observed.

The heatmap illustrates the coefficients of the Pearson correlation between differential metabolites in chili plants and soil bacterial diversity indices for two sets of treatment comparisons: C versus BC (left) and B versus BC (right). The saturation of the color of each cell represents the magnitude of the correlation coefficient, with the blue color indicating negative correlations, the red color denoting positive correlations, and white signifying non-significant correlations. A high absolute value of the Pearson correlation coefficient suggests a strong linear relationship between the variables. The heatmap shows the correlations between the top 20 most variable metabolites and soil bacterial diversity indices.

## 3. Discussion

### 3.1. Impact of Biochar on Soil Properties

The targeted soil analysis revealed that the application of biochar alone resulted in a significant increase in soil carbon and available phosphorus, a finding of paramount importance for the enhancement of nutrient availability for plant growth. A longitudinal study has underscored the role of soil carbon accumulation in elevating the rhizosphere microbial community, as elucidated in the work of Tian et al. [15], which highlights the microbially mediated mechanisms underlying the soil carbon accrual fostered by conservation agriculture under prolonged warming. The significance of phosphorus in plant development is manifold, with the dynamics of available phosphorus in the soil being intricately linked to factors such as the soil pH, aggregate structure, and elemental composition, as reviewed by Jindo et al. [16].

Our detection results indicate that the direct replenishment of the soil’s elemental components through the addition of coconut shell biochar notably enhanced the availability of plant nutrients. There was a slight decrease in the pH value, which, while not statistically significant, hinted at the positive impact of biochar on the soil’s physical structure. This partly explains the observed increase in available phosphorus in the soil treated with biochar. The correlation between the rise in the soil carbon content and available phosphorus and the level of plant growth in this study aligns with previous research, such as [17], which demonstrates the positive influence of biochar on soil carbon and phosphorus levels. Furthermore, studies on soil amendment for crops [18] also acknowledge the fertilizing effects of biochar when applied independently. Although these studies do not directly link soil nutrient replenishment to microbial communities, they support our observations and lay the groundwork for understanding the subsequent changes in microbial communities and plant metabolic differences within the soil.

Research also indicates that the impact of biochar application on soil carbon sequestration and plant nutrition is subject to structural and functional changes over time, an issue known as biochar aging, as analyzed by Meng et al. [19]. However, for short-cycle vegetable cultivation, such as chili pepper cultivation, the issue of biochar aging is largely circumvented.

### 3.2. Impact of Bacillus Strain Ya-1 on Microbial Activity

The standalone application of *Bacillus* strain Ya-1 was found to increase soil microbial biomass nitrogen (N) and phosphorus (P), highlighting the strain’s role in enhancing microbial activity. This activity is essential for nutrient cycling and for suppressing pathogenic microorganisms through competitive exclusion and the production of antimicrobial compounds. Our findings echo those in [20,21], which also emphasized the strain’s role in activating plant roots and promoting the recruitment of beneficial microbial populations. Furthermore, [22] discussed the mechanism by which antagonistic bacteria can enhance plants’ resistance to soil-borne pathogens by activating plant immune pathways through root colonization.

### 3.3. Synergistic Effects of Biochar and Bacillus Strain Ya-1

When biochar and *Bacillus* strain Ya-1 were applied together, a synergistic enhancement of soil nitrogen content, pH, and microbial biomass carbon was observed. This suggests an interaction between the physical properties of biochar and the biological functions of *Bacillus* Ya-1. Study [23] has shown that the microstructure of biochar can positively influence microbial habitation, particularly in the formation of biofilms by plant growth-promoting rhizobacteria (PGPR). Infrared spectroscopy and community diversity data from [24] further demonstrated the recruitment of beneficial microorganisms by different surface functional groups on biochar. The provision of a conducive habitat by biochar likely enhances the effectiveness of rhizosphere-dwelling strains like *Bacillus* Ya-1 [25]. The integration of biochar and *Bacillus* strain Ya-1 also significantly impacted rhizosphere bacterial diversity, as evidenced by positive effects on Pielou’s evenness and Shannon diversity indices, which are critical determinants of a healthy and resilient microbial community.

### 3.4. Enhancement of Microbial Diversity and Soil Health

The integration of biochar with microbial inoculants, such as *Bacillus* strain Ya-1, has been demonstrated to significantly enhance microbial diversity and soil health, contributing to improved plant nutrient uptake and reduced susceptibility to soil-borne diseases [26]. Biochar’s role in amending soil and facilitating microbial activity is particularly notable, as it can alter microbial habitats and interactions, thereby enhancing the efficacy of biological control agents [27]. Recent studies have highlighted the potential synergistic effect of biochar in combination with microbial agents for improving soil health and plant growth. For instance, the addition of biochar to wheat–maize systems have been shown to stimulate microbial richness and enzyme activities, leading to increased soil organic carbon and total nitrogen [28]. Moreover, modified biochar has been effective in enhancing soil fertility and rice yield in mercury-contaminated soils, underscoring its role in mitigating the impacts of soil pollution [29]. The combined application of biochar with bacteria and plants has also shown promise in the remediation of oil-contaminated soils, indicating the broad application potential of this approach [30].

Furthermore, the interaction between biochar, soil, and plants is crucial for sustainable agriculture under changing climate conditions [31]. The synergistic effect of biochar and microorganisms has been shown to greatly improve the vegetation and microbial structure in degraded alpine grasslands, contributing to the restoration of these fragile ecosystems [32]. These findings underscore the importance of biochar and microbial interactions in enhancing soil health and microbial diversity, which is essential for maintaining soil fertility and supporting plant growth.

### 3.5. Metabolite Regulation of Plant Growth

In this study, we observed that key metabolites, particularly cinnamic acid and its derivatives, significantly promoted the growth of chili plants under the combined application of biochar and *Bacillus* strain Ya-1. As a natural plant growth-promoting compound, cinnamic acid has been demonstrated to enhance plant growth by modulating phytohormone signaling and strengthening the antioxidant defense system [33]. In this research, the notable increase in the cinnamic acid levels under the BC treatment was closely correlated with the improvement in plant growth parameters. Furthermore, the antioxidant and pro-oxidant activities of cinnamic acid derivatives within the plant body are considered key factors in regulating plant cell growth [34], affecting cell division and elongation by influencing the levels of reactive oxygen species (ROS) and the activity of antioxidant enzymes.

Moreover, cinnamic acid and its derivatives also play a significant role in the plant’s response to environmental stress. These metabolites are involved in intracellular signaling pathways, aiding plants in adapting to variable environmental conditions [35]. Under the conditions of combined biochar and microbial application, these metabolites may enhance the antioxidant system of chili plants, improve the plants’ tolerance to harmful substances in the soil, and, thus, promote healthy plant growth.

## 4. Materials and Methods

### 4.1. Pot Experiment

#### 4.1.1. Experimental Groups

The pot experiment was carried out in a room under controlled temperature and light conditions and lasted 52 days. “Boyou” (*Capsicum annuum* Var. *longum*) was the variety of pepper used in the present study. The seeds were pre-germinated on moist filter paper in a dark environment at 25 °C, and then seedlings with uniform germination were transplanted into plastic pots (10 cm in diameter and 12.5 cm in height) containing 0.95 L of soil. The soil used in the experiment was collected from the experimental field of the Hainan Academy of Agricultural Sciences (110.37° E, 20.01° N), followed by its crushing and the removal of stones and debris. The biochar used was produced via the pyrolysis of coconut coir at 650 °C for 30 min and had a pH value of 8.16. The pot experiment consisted of four treatment groups, including the control group (CK), bacterial inoculant treatment group (B), biochar treatment group (C), and biochar–bacteria treatment group (BC), with three biological replicates per treatment group. For the C and BC groups, biochar and soil were uniformly mixed at a mass ratio of 2%. During the experiment, the temperature was set to 25 °C, with a 16 h light/8 h dark cycle. The watering of the plants was performed every 2–3 days to ensure that the soil moisture was maintained above 30%, and it ceased three days before the end of the experiment.

#### 4.1.2. Preparation of the Inoculum of the Biochar-Immobilized *Bacillus* Strain Ya-1

The previously isolated *Bacillus* strain Ya-1 used in this study was kept in our laboratory [15,36,37]. A single colony of bacterium grown on the plate was inoculated into 100 mL of the liquid LB medium containing 1 g NaCl, 1 g tryptone, and 0.5 g yeast extract dissolved in 100 mL of water and incubated at 28 °C for 24 h with constant shaking at 200 rpm to obtain the pure culture of this bacterial strain. The concentration was confirmed by measuring the OD600 of the bacterial culture solution with a spectrophotometer, and fresh LB medium was used to adjust the bacterial solution to a concentration of 10^8^ cfu/mL. Thereafter, 20 mL of this culture inoculum was added to a flask containing 200 mL of fresh LB liquid medium and incubated at 28 °C with a shaking speed of 150 rpm. The flask received 20 g of biochar every 8 h, with the pH value being maintained below 7.5 by using 0.1 mol/L of the NaOH solution and dilute hydrochloric acid. The cultures were incubated for 48 h, and to remove the extra moisture, the inoculum was air-dried.

### 4.2. The Measurement of Soil Edaphic Factors

After carrying out the pot experiment, the careful removal of soil surrounding the plant roots using a brush was performed, followed by the discarding of plant tissues. The methods used for the determination of the physicochemical properties of the soil are listed in Table 1.

### 4.3. The Investigation of Bacterial Diversity via 16S Amplicon Sequencing

The TGuide S96 Magnetic Soil/Stool DNA Kit (Tiangen Biotech (Beijing) Co., Ltd., Beijing, China) was used to extract the soil’s total DNA. The extracted DNA served as a template for the amplification of 16S rDNA with the primers, including 338F (5′-ACTCCTACGGGAGGCAGCA-3′) and 806R (5′-GGACTACHVGGGTWTCTAAT-3′) [38] using a PCR machine. The PCR products resulting from amplification were collected and sent to Beijing Tsingke Biotech Co., Ltd., Guangzhou Sequencing Centre (Guangzhou, China), and then they were purified, quantified, normalized, and used for library construction for next-generation sequencing. Qualified libraries were then subjected to the Illumina NovaSeq 6000 platform (Illumina, Inc., San Diego, CA, USA) for sequencing.

Trimmomatic v0.33 was used to filter raw sequencing reads, and thereafter, the identification and removal of primer sequences were performed using cutadapt v1.9.1, which resulted in the generation of clean reads that did not contain primer sequences. The DADA2 method (divisive amplicon denoising algorithm) [39] implemented in QIIME2 2020.6 [40] was employed to perform denoising. To obtain the final valid data (non-chimeric reads), the first paired-end sequencing reads were merged, and then chimeric sequences were identified and removed. The analysis of diversity was performed on the sequencing data.

### 4.4. The Measurement of the Agronomic Traits of Chili Plants

After carefully removing the plants from the soil using a brush, they were thoroughly cleaned. The entire plant, as well as the separated individual roots, stems, and leaves, was photographed. The determination of the plant height and leaf surface area was performed using MATO V 4.2 software [41]. The roots were stained using the triphenyl tetrazolium chloride (TTC) reduction method, and their vitality was assessed.

### 4.5. The Extraction of Metabolites and LC-MS/MS Analysis

The fresh aerial parts of plants from each treatment group were harvested, transferred into centrifuge tubes, and quenched in liquid nitrogen. The plant samples were freeze-dried and ground, and 25 mg of each was then weighed into EP tubes at a low temperature. Homogenizer beads and 1000 μL of the extraction solution, a mixture of methanol/acetonitrile/water at a ratio of 2:2:1 *v*/*v*, which contains an isotope-labeled internal standard, were together added to the tubes. The samples were vortexed for 30 s, followed by homogenization using a homogenizer operated at 35 Hz for 4 min and the ultrasonic extraction in an ice-water bath for 5 min three times. The samples were then allowed to stand for 1 h at −40 °C, and next, they were centrifuged at 4 °C and 12,000 rpm with a relative centrifugal force of 13,800× *g* and rotational radius of 8.6 cm for 15 min. The resulting supernatant was collected and centrifuged again at 4 °C and 12,000 rpm for 15 min with the same conditions. Finally, the supernatant was transferred into new centrifuge vials for LC-MS/MS analysis, and an equal volume of it was obtained from all samples and was mixed and subjected to quality control (QC) testing using the LC-MS/MS equipment (Thermo Fisher Scientific, Waltham, MA, USA).

To analyze the non-polar metabolites, a Thermo Fisher Scientific Vanquish ultra-high-performance liquid chromatography (UHPLC) system (Thermo Fisher Scientific, Waltham, MA, USA) was used, which allows for the chromatographic separation of target compounds with a liquid chromatography (LC) column (Phenomenex Kinetex 2.6 μm C18 (2.1 mm × 50 mm). To acquire primary-ion and secondary-ion mass spectrometry data, we employed an Orbitrap Exploris 120 mass spectrometer (Thermo Fisher Scientific, Waltham, MA, USA). The instrumental parameters of UHPLC included a sheath gas flow rate of 50 Arb, an auxiliary gas flow rate of 15 Arb, a capillary temperature of 320 °C, and spray voltages of 3.8 kV (positive) and −3.4 kV (negative). The collected raw data were subjected to filtration and standardization, followed by annotation, screening, and analysis by comparison with the database.

### 4.6. Statistical Analysis Methods

The measurement results of the soil pH, moisture content, elemental content, and other parameters were organized, and the Log2FC of each group’s parameter changes were calculated based on the CK group, followed by the creation of a heatmap. The diversity index was calculated based on the results of microbial diversity sequencing, and the Kruskal–Wallis test was used to compare the diversity indices between groups. Tukey’s Honest Significant Difference was used for multiple comparisons and the ranking of means (*p* < 0.05). The collected plant growth parameters were organized, and ANOVA was used to compare the mean differences of parameters between groups (*p* < 0.05). All tests in this study were calculated using Rstudio (Posit Software, Boston, MA, USA) based on R version 4.4.1, using the packages pheatmap, vegan, agricolae, and ggplot2.

## 5. Conclusions

This study provides compelling evidence that the integration of coconut shell biochar and *Bacillus* strain Ya-1 significantly enhances soil quality and microbial diversity, concurrently promoting the growth of chili plants. The combination leverages the unique properties of biochar and the biological control potential of Ya-1, offering synergistic benefits for soil health and plant performance. This strategy could substantially reduce our reliance on chemical inputs by improving soil properties and bolstering microbial activity, thereby contributing to sustainable agricultural practices.

The notable increases in soil nutrient content, along with improvements in microbial diversity and plant growth parameters, position the integrated application of biochar and Ya-1 as a promising approach for sustainable agriculture. These benefits are especially significant in regions susceptible to soil degradation and intensive chemical use, where sustainable management practices are essential for long-term productivity. The treatment’s profound impact on plant physiology is further indicated by metabolic changes that may enhance crop resilience and yield. While this research offers a comprehensive understanding of the effects of biochar and bacterial strains on chili plants under controlled conditions, the translation of these findings to field applications necessitates further investigation. This includes addressing the challenges of scalability and economic viability.

Future research should focus on the long-term impacts of biochar and *Bacillus* treatments across various soil types and climates, ensuring the universality of their benefits. A deeper exploration of plant–microbe interactions in the rhizosphere will elucidate the mechanisms behind the enhancement of soil health and plant growth through the integrated use of biochar and *Bacillus* strains. Such studies will inform the scaling up of this approach and its integration into broader sustainable agricultural practices.

In summary, the combined application of biochar and microbial agents emerges as a viable strategy to improve soil and plant health, laying the groundwork for developing more diversified and resilient agricultural systems. This approach holds potential for addressing global challenges such as climate change and food security.

## Figures and Tables

**Figure 1 ijms-25-11231-f001:**
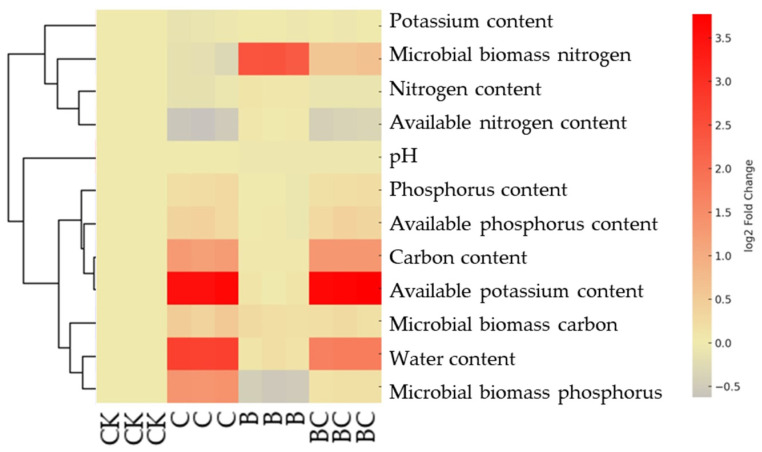
Heatmap analysis of the relationships between soil physicochemical properties and microbial biomass in different treatments.

**Figure 2 ijms-25-11231-f002:**
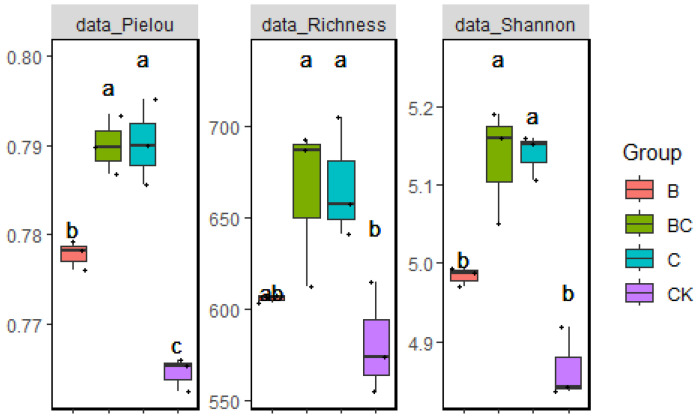
Boxplots of bacterial diversity indices in the rhizosphere soil. The boxplots represent the effects of different treatments, including the control (CK), biochar alone (C), Ya-1 alone (B), and biochar combined with Ya-1 (BC), on bacterial diversity indices (Pielou’s evenness, species richness, and Shannon diversity index) in the rhizosphere soil. Different letters above the boxes denote significant differences (*p* < 0.05) between treatment groups, ^+^
*p* ≤ 0.05.

**Figure 3 ijms-25-11231-f003:**
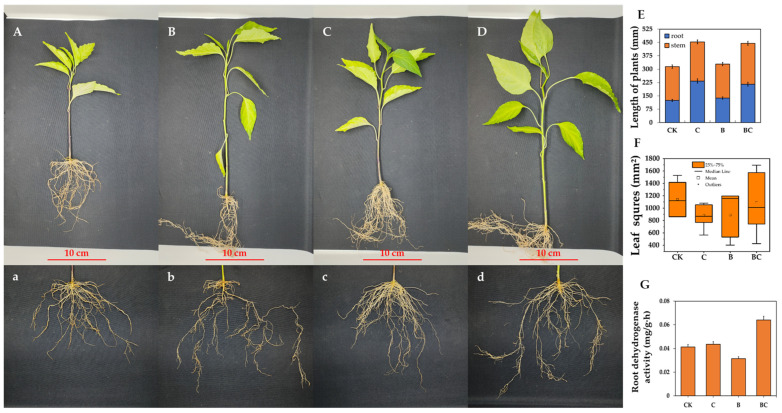
Plant growth pot illustrations and parameters in different treatments. (**A**) Full plant of control (CK); (**B**) full plant of treatment biochar application alone (**C**); (**C**) full plant of Ya-1 application alone (**B**); (**D**) full plant of biochar co-applied with Ya-1 (BC); (**E**) bar graph showing the total height of chili plants, including the root and stem lengths, in different treatments, including the control (CK), biochar application alone (**C**), Ya-1 application alone (**B**), and biochar co-applied with Ya-1 (BC); error bars represent the standard error of the mean. (**F**) The boxplot of the leaf surface area of chili plants subjected to different treatments; in the boxplot, the box shows the interquartile range (IQR) of the data, while the line inside the box represents the median values, and whiskers indicate the range; dots denote the outliers. (**G**) Bar graph for the dehydrogenase activity of the roots of chili plants grown under different treatments; error bars represent the standard error of the mean (*p* < 0.05). (**a**) Underground part of control (CK); (**b**) underground part of biochar application alone (**C**); (**c**) underground part of Ya-1 application alone (**B**); (**d**) underground part of biochar co-applied with Ya-1 (BC).

**Figure 4 ijms-25-11231-f004:**
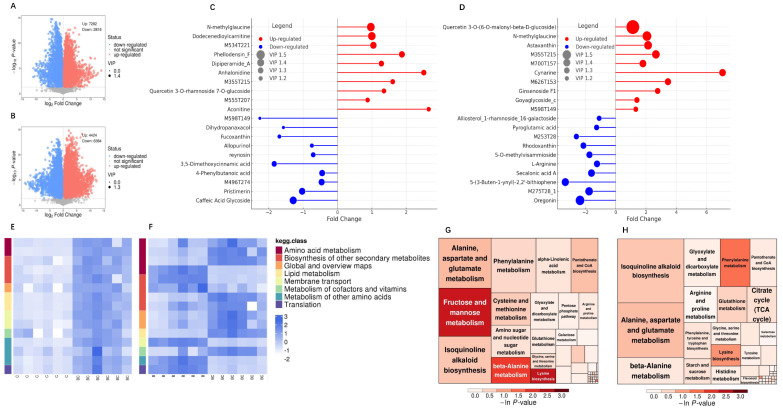
Metabolic profiling and KEGG pathway enrichment analysis of chili plants grown under different treatments. (**A**) The volcano plot of differential metabolites between the treatment C and the treatment BC; (**B**) the volcano plot of differential metabolites between the B and BC treatments; (**C**) the matchstick diagram showing the top 10 differential metabolites between the C and BC treatments; (**D**) matchstick plot of the top 10 differential metabolites between the B and BC treatments; (**E**) heatmap of the KEGG pathway enrichment analysis of differential metabolites between the C and BC treatments; (**F**) heatmap of differential metabolites between the B and BC treatments annotated by KEGG; (**G**) treemap showing the KEGG pathway enrichment analysis of differential metabolites between the treatment C and the treatment BC; (**H**) treemap of differential metabolites between the B and BC treatments as detected by the KEGG pathway enrichment analysis.

**Figure 5 ijms-25-11231-f005:**
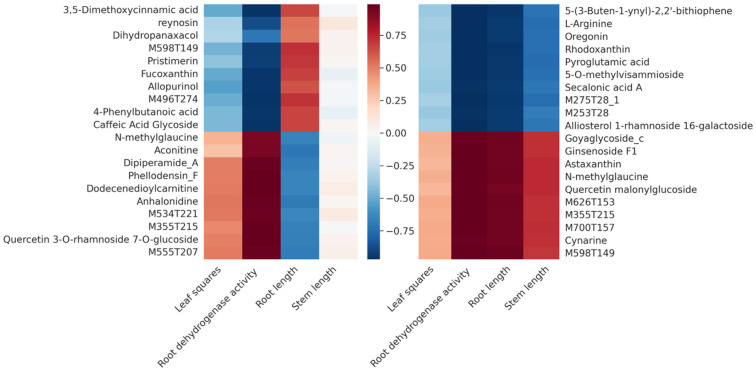
The heatmap of the Pearson correlation between differential metabolites and phenotypic parameters for growth.

**Figure 6 ijms-25-11231-f006:**
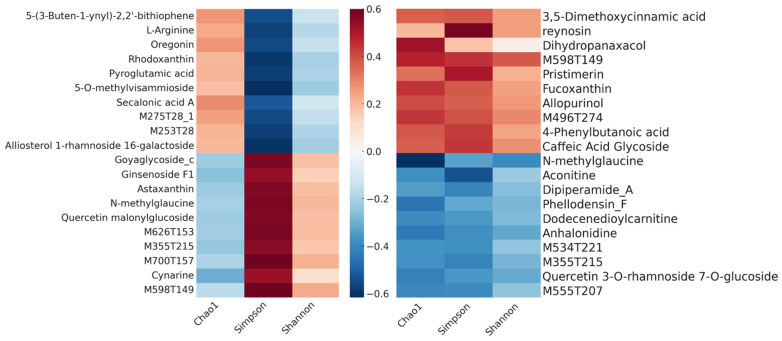
The heatmap of the Pearson correlation between differential metabolites and statistical parameters for the biodiversity indices of soil bacterial community.

**Table 1 ijms-25-11231-t001:** Methods used for the analysis of soil physicochemical properties.

Detection Index	Detection Method
pH	Electrode pH meter
Water content	Oven
Carbon	Elemental analyzer
Nitrogen	Elemental analyzer
Quick-acting nitrogen	Alkaline diffusion
Quick-acting phosphorus	Molybdenum antimony colorimetry
Quick-acting potassium	Flame photometric
Microbial carbon	Chloroform fumigation and leaching
Microbial nitrogen	Chloroform fumigation and extraction
Microbial phosphorus	Chloroform fumigation and extraction
Phosphorus, potassium, sodium, calcium, magnesium, sulfur, iron, manganese, copper, zinc, molybdenum, and boron	ICP-MS/ICP-OES

Note: ICP-MS stands for “Inductively Coupled Plasma-Mass Spectrometry”, while ICP-OES stands for “Inductively Coupled Plasma-Optical Emission Spectrometry”.

## Data Availability

Microbial diversity sequencing data (SAMN44111606, SAMN44111607, SAMN44111608, SAMN44111609) can be found at the following links: https://www.ncbi.nlm.nih.gov/biosample/44111606 (accessed on 9 October 2024); https://www.ncbi.nlm.nih.gov/biosample/44111607 (accessed on 9 October 2024); https://www.ncbi.nlm.nih.gov/biosample/44111608 (accessed on 9 October 2024); https://www.ncbi.nlm.nih.gov/biosample/44111609 (accessed on 9 October 2024). The metabolite data can be found on MetaboLights as MTBLS11042, at https://www.ebi.ac.uk/metabolights/editor/study/MTBLS11042 (accessed on 6 September 2024).

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
