# Peer review of "Enhancing the Growth of Chili Plants and Soil Health: Synergistic Effects of Coconut Shell Biochar and Bacillus sp. Strain Ya-1 on Rhizosphere Microecology and Plant Metabolism"

_ijms, 2024, doi:10.3390/ijms252011231_

Round 1
Reviewer 1 Report
Comments and Suggestions for Authors
Comments and Suggestions for Authors
Recently, there is a growing popularity of the application of biological control agents, as well as soil amendments like biochar. The manuscript entitled " Enhancing the Growth of Chili Plant and Soil Health: Synergistic Effects of Coconut Shell Biochar and Bacillus sp. strain Ya-1 3 on the Rhizosphere Microecology and Plant Metabolism " examined the joint effects of coconut shell biochar and Bacillus strain Ya-1 on soil fertility, rhizosphere bacterial communities, and the growth of chili.
The study is very interesting, especially the discovery that cinnamic acid and its derivatives can significantly promote the growth of chili peppers when used in combination with bacteria and biochar is innovative. The problems with the work are as follows,
1. This manuscript assumed that synergistic effects can significantly improve rhizosphere microbiota, thereby enhancing soil fertility, increasing microbial diversity, and improving the growth and productivity of chili plants. But why the authors had such a hypothesis is not clearly explained in the article.
2. The methods of the paper were not adequately described. For example, 1) The study lasted for 52 days. But is that starting from sowing or not? Is there a difference in germination rate and emergence time between different treatments? 2) The primer information in the research method needs to be supported by literature. 3) Is there any relevant data on the element content of coconut shell biochar?
3. In the results, there were significant differences in soil physicochemical properties between different treatments, such as contents of total P (line170). Why?
4. In Figure1, the control group (CK) was used to establish a baseline for all evaluated parameters, and then should it be marked as the lightest color 0?
5. In Figure2, there was no significant difference in the three diversity indices of chili rhizosphere bacteria between BC and C, so that the BC group exhibited the maximum Shannon diversity index was not rigorous(line 190) .
6. In addition, the image numbering in this paper is confusing.
Author Response
Comments 1: This manuscript assumed that synergistic effects can significantly improve rhizosphere microbiota, thereby enhancing soil fertility, increasing microbial diversity, and improving the growth and productivity of chili plants. But why the authors had such a hypothesis is not clearly explained in the article.
Response 1: Thank you for your comments. We agree that the original expression was indeed not smooth enough. Now we have reorganized the introduction section, adding a literature review on the basic mechanisms of biochar soil amendment and synergistic microorganisms from line 54 to 57, and stating the basic idea and expected purpose of the study design at line 78. We have removed the description of the hypothesis, which is to reduce arbitrary inferences. Now we believe that the new paragraph avoids weak logical inferences and is more in line with our original intent.
Comments 2: The methods of the paper were not adequately described. For example, 1) The study lasted for 52 days. But is that starting from sowing or not? Is there a difference in germination rate and emergence time between different treatments? 2) The primer information in the research method needs to be supported by literature. 3) Is there any relevant data on the element content of coconut shell biochar?
Response 2: 1) It has come to our attention that the experimental timeline was not clearly defined. We would like to clarify that the period of seed germination was excluded from the experimental duration. We selected seedlings that had simultaneously sprouted and successfully developed both shoots and roots from a large batch of pre-germinated seeds for transplantation. This selection process ensured uniformity among the seedlings used in the experiment. The experiment's timeline commenced upon the transplantation of these seedlings. We have added this point in lines 95-96.
2) We appreciate the prompt to elaborate on the primers used. In response, we have now included more detailed information about the primers in our manuscript, along with pertinent references for further reading.
3) Our research on this particular coconut charcoal process is still in its early stages. Currently, we are in the process of conducting SEM and FTIR analyses. We apologize for any inconvenience, but we are unable to provide additional data at this time. We will, however, include these results in our forthcoming studies.
Comments 3: In the results, there were significant differences in soil physicochemical properties between different treatments, such as contents of total P (line170). Why?
Response 3: Yes, you have identified a noteworthy point; in fact, we attempted to explain this phenomenon in Section 4.1 of the discussion. We hope that the rewritten Section 4.1 clarifies the issue.
Comments 4: In Figure1, the control group (CK) was used to establish a baseline for all evaluated parameters, and then should it be marked as the lightest color 0?
Response 4: We fully understand your concerns regarding the baseline of the heatmap. After some experimentation, Figure 1 now uses the values from the CK group of each parameter as the baseline for plotting. However, we are not certain if this method of presentation is more readable. Should you have additional suggestions regarding the color scheme, please do let us know so we can make the necessary adjustments.
Comments 5: In Figure2, there was no significant difference in the three diversity indices of chili rhizosphere bacteria between BC and C, so that the BC group exhibited the maximum Shannon diversity index was not rigorous (line 190).
Response 5: Agree. We have meticulously revised the descriptions according to the data and images, and have also included additional appropriate explanations, which can be found from line 226 to line 232.
Comments 6: In addition, the image numbering in this paper is confusing.
Response 6: Yes, thank you for pointing this out. It was indeed a careless oversight on our part. We have now reviewed all the numbers.

Reviewer 2 Report
Comments and Suggestions for Authors
The manuscript entitled "Enhancing the Growth of Chili Plant and Soil Health: Synergistic Effects of Coconut Shell Biochar and Bacillus sp. strain Ya-1 on the Rhizosphere Microecology and Plant Metabolism" is an interesting article discussing a study aimed at reducing soil degradation and decreasing reliance on chemical inputs in agriculture. The research focused on the combined effects of coconut husk biochar and the Bacillus strain Ya-1 on soil fertility, rhizosphere bacterial communities and growth of chilli plants (Capsicum annuum L.). The content presented in the manuscript fits well within the thematic scope of the IJMS. The information presented in the manuscript will certainly be of interest to potential readers.
The manuscript is well-structured. It contains documentation: 6 figures and 1 table.
I have only a few suggestions for the authors before publishing this manuscript:
1. The introduction chapter is rather superficial. Please describe in more detail, based on the most recent literature, the effect of biochar on soil quality: physical, chemical, and biological properties, and on plant growth and development.
2. The chapter "Materials and Methods" needs to be expanded. Specify the characteristics of the soil used in the experiment, what was its granulometric composition, what was the content of organic carbon, nitrogen, hydrolytic acidity, degree of soil saturation with base cations. How much soil was in a pot. What was the soil moisture during the pot experiment. It is also necessary to indicate the number of bacteria introduced per 1 kg of soil. Was the density of the bacteria measured in cm3? In the chapter "2. Materials and methods" there should be a subsection on statistical calculations, where you should write which methods you used to statistically verify the results obtained. Were the bacterial sequences obtained deposited in an international database, e.g. NCBI, available to potential readers?
3. In the "Results" or "Supplementary materials" section, please provide a photograph documenting the pot experiment.
4. The Discussion section, especially subsection "4.1. Effect of biochar on soil properties" needs more in-depth analysis. Please describe in more detail how biochar changes soil properties and what the mechanism is in chemical terms.
Author Response
Comments 1: The introduction chapter is rather superficial. Please describe in more detail, based on the most recent literature, the effect of biochar on soil quality: physical, chemical, and biological properties, and on plant growth and development.
Response 1: We greatly appreciate your insightful suggestions. In response, we have incorporated additional literature and restructured the sections pertaining to the physicochemical properties of biochar, its functional mechanisms, and the integration of biochar with microorganisms. These revisions significantly enhance the strength and clarity of the presentation. The key updates commence from line 54, and we believe they address your concerns effectively. We are hopeful that the revised section will meet with your approval.
Comments 2: The chapter "Materials and Methods" needs to be expanded. Specify the characteristics of the soil used in the experiment, what was its granulometric composition, what was the content of organic carbon, nitrogen, hydrolytic acidity, degree of soil saturation with base cations. How much soil was in a pot. What was the soil moisture during the pot experiment. It is also necessary to indicate the number of bacteria introduced per 1 kg of soil. Was the density of the bacteria measured in cm3? In the chapter "2. Materials and methods" there should be a subsection on statistical calculations, where you should write which methods you used to statistically verify the results obtained. Were the bacterial sequences obtained deposited in an international database, e.g. NCBI, available to potential readers?
Response 2:
1) We have measured the basic components of the soil in Section 2.2 and described the results in Section 3.1.1, where you can find them in the CK group data.
2) Lines 95-98, we have added information about the seed germination treatment and the volume of soil.
3) Lines 106-108, we have included details on soil moisture management. It should be noted that ceasing irrigation a few days before sampling was to measure the water retention capacity of the potted soil under different treatments.
4) Regarding the measurement of bacterial concentration, we have added information in Section 2.1.2, lines 114-117.
5) We have added a specific section on statistical methods in 2.6.
6) The microbial diversity data has now been uploaded, and we have included additional information in the Data Availability Statement section following the main text. Thank you for your reminder.
Comments 3: In the "Results" or "Supplementary materials" section, please provide a photograph documenting the pot experiment.
Response 3: Agree. In the “Results”, figure 3 now includes images of the plants.
Comments 4: The Discussion section, especially subsection "4.1. Effect of biochar on soil properties" needs more in-depth analysis. Please describe in more detail how biochar changes soil properties and what the mechanism is in chemical terms.
Response 4: Thank you for your suggestion. We have rewrite Section 4.1, incorporating additional literature review and reorganizing our interpretation of the phenomenon of biochar affecting soil composition changes. It has been significantly enhanced. We hope that the new Discussion 4.1 will meet your satisfaction.

Round 2
Reviewer 2 Report
Comments and Suggestions for Authors
I appreciate the authors' efforts to improve the manuscript. I accept the current version of the paper for printing.